# COVID-19 Long-Term Effects: Is There an Impact on the Simple Reaction Time and Alternative-Forced Choice on Recovered Patients?

**DOI:** 10.3390/brainsci12091258

**Published:** 2022-09-16

**Authors:** Mauro Santoyo-Mora, Carlos Villaseñor-Mora, Luz M. Cardona-Torres, Juan J. Martínez-Nolasco, Alejandro I. Barranco-Gutiérrez, José A. Padilla-Medina, Micael Gerardo Bravo-Sánchez

**Affiliations:** 1Division of Postgraduate Studies and Research, Tecnológico Nacional de México en Celaya, Celaya 38010, Mexico; 2Division of Sciences and Engineering, Universidad de Guanajuato Campus León, León 37150, Mexico; 3Department of Education and Research in Health, Hospital General Zona 4 Instituto Mexicano del Seguro Social, Celaya 38060, Mexico; 4Department of Mechatronics Engineering, Tecnológico Nacional de México en Celaya, Celaya 38010, Mexico; 5Department of Electronics Engineering, Tecnológico Nacional de México en Celaya, Celaya 38010, Mexico; 6Department of Biochemistry Engineering, Tecnológico Nacional de México en Celaya, Celaya 38010, Mexico

**Keywords:** choice behavior, COVID-19, decision making, psychophysics, reaction time

## Abstract

A comparative single-evaluation cross-sectional study was performed to evaluate cognitive damage in post-COVID-19 patients. The psychophysics tests of Two-Alternative Forced Choice (2AFC) and Simple Reaction Time (SRT), under a designed virtual environment, were used to evaluate the cognitive processes of decision-making, visual attention, and information processing speed. The population under study consisted of 147 individuals, 38 controls, and 109 post-COVID patients. During the 2AFC test, an Emotiv EPOC+^®^ headset was used to obtain EEG signals to evaluate their Focus, Interest, and Engagement metrics. Results indicate that compared to healthy patients or recovered patients from mild-moderate COVID-19 infection, patients who recovered from a severe-critical COVID infection showed a poor performance in different cognitive tests: decision-making tasks required higher visual sensitivity (*p* = 0.002), Focus (*p* = 0.01) and information processing speed (*p* < 0.001). These results signal that the damage caused by the coronavirus on the central nervous and visual systems significantly reduces the cognitive processes capabilities, resulting in a prevalent deficit of 42.42% in information processing speed for mild-moderate cases, 46.15% for decision-making based on visual sensitivity, and 62.16% in information processing speed for severe-critical cases. A psychological follow-up for patients recovering from COVID-19 is recommended based on our findings.

## 1. Introduction

After two plus years of the COVID-19 pandemic, an important impact on the global public health has been reported, with over 518 million confirmed cases and over six million deaths, according to the report by the World Health Organization of 15 May 2022 [1]. Health researchers have found that, while most of the infected population has been asymptomatic or has had mild symptoms with quick recovery, a fraction of the infected population has undergone several long-term symptoms [2]. Lopez-Leon et al. estimate that around 80% of the infected patients have developed at least one long-term symptom [3], mainly manifested through the peripheral organs, the central and peripheral nervous systems (CNS and PNS respectively), the ocular system, and the cardiovascular and respiratory systems [2,3,4].

Several healthcare organizations began to report COVID-19-related neurological disorders based on cases around the globe and started to hypothesize a set of possible consequences derived from these disorders [5,6,7,8,9]. Neurological manifestations found in the CNS and PNS show different grades of severity after a critical COVID infection [8]. These manifestations were mainly reported as cerebrovascular events (strokes), encephalopathies, encephalitis, confusion, and delirium, and in some cases, Guillain-Barré syndrome [9]. In these cases, the use of electroencephalography (EEG) allows for identifying different abnormalities in the functioning of the CNS, opening the possibility of developing an earlier diagnosis of neurological impairments in critical patients, as well as a prompt proposal of adequate treatments after the infection [10,11].

Related to COVID-19 infection, there exists evidence of a rise in the disruption of the visual system [12,13]. Most of the common disruptions found are conjunctivitis, ocular pain, diplopia, visual impairment, eye dryness, red eyes, changes in the intraocular pressure (IOP), retinal changes, blurry vision, and light sensitivity [4,12,14,15]. On the other hand, there is evidence that when the infection of COVID-19 is accompanied by severe respiratory distress, it produces a significant impairment in certain cognitive processes, such as visual memory tasks, perception, and reasoning [16,17]. Furthermore, a decrease in the information processing speed, strongly related to reaction times, can be used as a risk estimator of death by COVID-19 [18]. Psychologists pay attention to the reaction time as a tool to evaluate the limitations on the nervous system functioning, becoming a referent for complex reasoning and knowledge evaluation [19].

The evaluation of mental abilities can be performed with standardized psychological tests or virtual technologies, as in the case of red light-green light reaction time [20]. The development and application of virtual environments open the possibility to create tools for the diagnosis, training, or treatment of some disorders. These virtual environments increment the similarities to the task and their scenarios, as could be the case of cognitive training for attention-deficit and hyperactivity disorder, autism spectrum disorder, or even the enhancement of cognitive performance [21,22,23]. The strength of the virtual environments arises from the “sense of presence” experienced by the user [23], which gives the possibility of approaching near “real-life behaviors” inside a laboratory.

Our aim in this work is to study the performance of post-COVID population in cognitive processes such as decision-making based on their visual sensitivity, visual attention, and information processing speed, with the hypothesis that the grade of severity of the COVID-19 infection is related to a reduction in performance. To test this hypothesis, we propose a single-evaluation comparative cross-sectional study applying Multiple Alternative Forced Choice (MAFC) and Simple Reaction Time tests designed under virtual environments. An Emotiv EPOC+ headset is used to verify the Focus, Interest, and Engagement performance during the MAFC task.

## 2. Materials and Methods

### 2.1. Study Design and Setting

The present comparative cross-sectional study was developed at Hospital General Zona 4 (HGZ-4), of the Instituto Mexicano del Seguro Social, located at Celaya, Guanajuato, México, from July 2021 to May 2022. The local Research Committee and the Ethics Committee in Research from this healthcare institution revised and approved the research protocol of the study. The protocol was presented with the title “Neurological comparison between healthy and post-COVID subjects using psychophysics experiments” and was approved with registration number R-2021-1008-014. The present study fulfilled all requirements for observational studies based on the STROBE guidelines.

The groups of participants consisted of insured patients from HGZ-4. The recruitment process began on 19 July 2021, was kept active during data collection, and concluded on 31 May 2022. The participants were grouped into three categories: one group of mild-moderate COVID patients (MM), one of several-critical patients (SC), and a control group (CT) of patients who were never diagnosed with COVID. In the case of COVID patients, their group assignment was defined based on the COVID-19 spectrums [24].

Once patients were selected, we invited them to attend a scheduled appointment where the study would be carried out under comfortable conditions, in a quiet room with adequate illumination. Before the study, all patients signed an informed consent form of participation. First, a clinician interviewed the patient to assess his or her medical history to confirm the assigned group category; then, a Mini-Mental State Examination (MMSE) was carried out by a psychologist, to identify possible issues related to cognitive impairments. Finally, the patient was seated in a comfortable straight position in front of the computer’s screen, for the Two-Forced Alternative Choice (2AFC) and Simple Reaction Time (SRT) tests [25], each one performed only once, and carried out under virtual environments. For the 2AFC test, a portable EEG headset was mounted on the patient to record the Focus, Interest, and Engagement signals.

In principle, the tests are very simple, and it should not make any difference if the patient had previous knowledge of the use of computers before taking them. To further ensure that this would not bias the study, all participants were exposed to a sensibilization and training stage before the evaluation, consisting of a previsualization of the procedure. Once participants verbally expressed their familiarity and understanding of the mechanics of the tests, they continued with the evaluations, where they were required to maintain focus on the test.

### 2.2. Participants

We recruited adult and middle-aged participants aged 19 to 64 years old, both never infected healthy participants and recovered patients from COVID-19. All participants were required to present a normal or corrected visual acuity of 20/20 based on the Snellen Eye chart. Participants scoring lower than 24 in the MMSE, under mental state altering medication, suffering of color vision defects, or reporting a previous sleeplessness night, were excluded from the study.

Individuals for the healthy control group were recruited directly at the HGZ-4. Here, the recruiter performed a first-screening process to verify that the candidates had not had a confirmed COVID diagnosis, not any type of chronic health condition, such as diabetes or hypertension.

We chose the post-COVID patients from the HGZ-4 Epidemiology department’s database. Subsequently, they were contacted by phone by a member of the hospital support staff, who gave each patient a detailed explanation of the aim of the study. Finally, if the patient decided to collaborate in the study, he or she was scheduled for an evaluation of their cognitive abilities at the health center.

### 2.3. Study Variables

The purpose of the 2AFC test is to evaluate the visual sensitivity and visual attention in a decision-making process, the individuals achieve an efficiency index (*d’*) computed with the number of correct responses (hits) and the number of incorrect responses (alarms); *d’* allows the evaluation of the sensitive adjustment in a decision-making threshold to discriminate relevant stimulus from noise background in a participant. The *d’* range is from 0 to 3.29, 0 for a participant with null efficiency with all responses wrong, and 3.29 for a participant that reaches perfection in the test [26,27].

For measuring Focus, Interest, and Engagement a range from 0 to 1 is used. Focus indicates the concentration or directed attention to a specific task, 0 indicates a non-concentration, and 1 indicates full concentration. Interest or Valence is defined as the aberrance to perform a task, and values near 1 represent the acceptance to perform the task. Engagement represents alertness and direct involvement in the task: values near 0 represent boredom, whereas values near 1 represent the perfect mixture of attention and concentration [28].

The SRT test is widely used to relate mental processes with physical responses, a measure of processing speed [25]. Our reaction time test measures the time between a visual stimulus is presented, in a virtual environment programmed, and the dominant foot response pushing a pedal. It allows us to indirectly evaluate the integrity of peripheral nerves involved in a perceptual visual task.

The outcomes for this study are the Efficiency Index *d’*; the performance metrics of Focus, Interest, and Engagement; and the reaction time. As the demographic variables (age, sex, and schooling) appear as potential confounders in our study, our tests were simply designed, so as to not represent a significant difficulty despite the age, sex, or schooling of the participants; i.e., there is no need of experience in computation. Namely, in the 2AFC test it is only necessary to move the mouse and select the road where the car was presented (see Figure 1), and for the SRT test the interaction is only pushing a pedal with the dominant foot. The restriction of cognitive processes might appear as a risk of bias.

### 2.4. Simple Reaction and Forced Choice Experiments

For evaluating the cognitive abilities of the participants, a virtual environment (VE) of two tests, 2AFC and SRT, was designed, with scenarios related to activities performed while driving a vehicle. These VEs have been created using Unity 2020.1.7f1 and made compatible 64-bit Windows platforms. A portable computer, processor Intel Core i7-9750H, 16GB of RAM, and a GPU NVIDIA GeForce GTX 1660Ti were selected.

The 2AFC test was based on the signal detection theory (SDT), involving the selection of one of two options for searching for a target in a noisy environment that hinders signal detection [26]. Our test presents a series of trials in which a car silhouette (target) is randomly placed either on the left or right road (two alternatives of choice) in a foggy environment (noise); an example of this is shown in Figure 1a. To begin the test, the participant clicks the start button, and immediately a car silhouette is presented for 300 ms. After this time, labels of “Izquierda” (left) and “Derecha” (right) appear (Figure 1b), and the participant must choose one of them by clicking. Then, a new trial begins.

The difficulty grade of the 2AFC tests is set by background noise factor (*f*), which allows for modifying fog density using Equation (1),
(1)$$f=\frac{1}{e^{{\left(D\times d\right)}^{2}}}$$
where d represents the depth (distance from the viewpoint and the drawn pixel in the scene), and D corresponds to the fog density [29]. The participant was trained with 20 trials of 2AFC with the fog (see Figure 1a, *f* = 0), to become familiarized with the procedure and to adopt a comfortable position; then, the 2AFC test in two consecutive phases of 200 trials each was undertaken. Phase one used *f* = 0.270 and phase 2 used *f* = 0.338. The exposition time and fog density were kept constant throughout the corresponding phase.

Efficiency Index (*d’*) is computed by correct (hits) and incorrect (alarms) responses using Equation (2):(2)$$d^{\prime }=\frac{1}{\sqrt{2}}\left[Z\left(H\right)-Z\left(F\right)\right]$$
where *Z*(*H*) is the number of hits’ *z*-score and *Z*(*F*) refers to the number of alarms’ *z*-score [27].

For M-alternative forced-choice experiments at SDT, a set of tables has been defined including *d’* value in the function of the observer’s percentage of correct responses (*P*(*c*)) to *M* alternatives [30]. Therefore, the participant’s *d’* was obtained by computing the *P*(*c*) of he/she, using *M* = 2.

To measure Focus, Interest, and Engagement in the 2AFC tests, a set of EEG signals was recorded and analyzed with an Emotiv EPOC+^®^ headset. Melnik et al. found that the Emotiv EPOC+^®^ can record EEG data during visual decision-making tasks with a comparable degree of confidence compared to research-grade EEG devices [31]. In addition, this portable EEG device has a simple setup for studies outside the laboratory. To record the EEG signals, the software EmotivPRO version 3.1.2.388 was used.

Emotiv EPOC+^®^ uses plated contact-sensors connected to felt pads fixed to plastic-arms in an arrangement based on the 10–20 system. The sensors are placed on the locations: AF3, F7, F3, FC5, T7, P7, O1, O2, P8, T8, FC6, F4, F8, and AF4, with two references at P3 and P4. This system also adds two alternative references at the left and right mastoids (M1 and M2 respectively). Each EEG channel has 14-bit resolution with 1 LSB of 0.51 μV. Sampling rate is configurable between 128 and 256 Hz [32,33]; here we used 128 Hz and the 5th order sinc notch filter at 50 Hz, achieving an effective bandwidth of 0.2–45 Hz.

The Simple Reaction Time (SRT) test was programmed using a virtual environment synchronized with a foot pedal (800Z-GL3, Allen Bradley, Milwaukee, WI, USA), which simulates a braking action when a bicycle randomly appears in the scene. Randomized appearance time is between 1 to 5 s and the bicycle has the same probability of appearing from right to left or vice versa. An example is shown in Figure 2.

The participant’s SRT time is computed by triggering the chronometer (timing by the CPU) once the bicycle appears in the scene, stopping it once the participant pushes the pedal. The participant was trained with 10 trials of the SRT to become familiarized with the pedal and to adopt a comfortable position; then, the SRT test was done with 200 targets to measure the reaction time.

### 2.5. Procedure

As mentioned above, the patient session consisted of four steps (see Figure 3). Firstly, the participant was interviewed about their clinical history and evaluated with the Mini-Mental State Examination as a measure of cognitive functioning [34]. Post-COVID patients were interviewed about their clinical history, type of COVID-19 test used for the diagnosis, symptomatology throughout the illness, and sequelae derived from the infection. Secondly, the EPOC+^®^ was set up, assuring a good EEG signal quality for all channels with EmotivPRO software. Thirdly, the participant remained relaxed for 3 min with open eyes to record his/her reference signals. Then, the procedure described for the 2AFC and SRT tests was carried out for each participant. It is important to note that the Emotiv headset was removed from the participant before the SRT test (see Figure 3, step 4). Individual participant data were saved for their posterior analysis.

### 2.6. Data Analysis

#### 2.6.1. Quantitative Variables

The decision-making data (number of hits and number of alarms) were preprocessed to identify repetitive patterns (e.g., consecutive alternate responses of left–right or vice versa). Participants with this pattern were considered inattentive and were excluded from the 2AFC tests; for the others, the *d’* index was computed.

Participants with disconnection or poor signal-to-noise ratio of key channels to compute Focus, Interest, and Engagement were discarded; a signal quality analysis to verify the recording and to ensure a quality signal of over 80% was made with the emotive software.

For SRT test trials the criteria to discard events were fortunate responses (times lower than 100 ms), missed target (the bicycle disappears and no response from the participant was received), and a stop time higher than 2000 ms of pressing the pedal.

#### 2.6.2. Statistical Methods

Throughout the study, the demographic variables (age, sex, and schooling) of the study groups were compared to ensure a balance between the groups, the samples were tested for normality with a Kolmogorov–Smirnov test and were compared with a *t*-test as independent groups with age, sex, and schooling as dependent variables. Additionally, a homogeneity test with Bartlett’s analysis was performed to verify that the age distribution had the same variance between the groups. For schooling, a Wilcoxon’s test with independent groups and the schooling years as a dependent variable was performed. A Levene’s test was used as a homogeneity test to corroborate the same variance in schooling years between the groups.

The data analysis consisted of three steps. First, the data were pre-processed to exclude the trials mentioned above in the Quantitative Variables Section. Second, the included data were analyzed with the Kolmogorov–Smirnov normality test. Finally, samples with parametric behavior were analyzed with a mean *t*-test, and samples with non-parametric behavior were analyzed with Wilcoxon’s test. Additionally, a Cohen’s *d* analysis was performed to measure effect size, and the influence of age and time since COVID-19 infection was adjusted with an intercept method age correction if necessary.

To assess whether demographic variables affect the results, an interaction analysis was performed for each test. The interactions were computed with Sex as a categorical variable, whose interaction effect is represented for the female, and the remaining variables Age and Schooling were treated as numerical, expressed in years. The interactions considered each confounder as independent and their possible relations were Sex × Age, Sex × Schooling, Age × Schooling. In addition, the post-COVID groups included the analysis for interactions between the sequelae derived from the infection, which included symptoms and disorders categorized as Respiratory, Cardiovascular, and Neurologic, as well as their interactions Respiratory × Cardiovascular, Respiratory × Neurologic, and Cardiovascular × Neurologic.

An analysis for the Severe-Critical group was included to observe the effects and differences of the COVID-19 infection between subgroups of adults (19 to 44 years) and the middle-aged adults (45 to 64 years); the comparison was performed with a Wilcoxon’s test and a Cohen’s *d* to identify the effect sizes. All the analysis in this study considered a significance value *α* = 0.05. The statistical analysis of these variables was performed with MATLAB R2016b (The MathWorks Inc., Natick, MA, USA).

## 3. Results

### 3.1. Study Participation

A total of 147 participants (51% males and 49% females) with a mean age of 38.7 years (*SD* = 11.3) completed the study. In total, 157 subjects attended the schedule; however, 10 did not qualify for eligibility due to being older than 64 years (*n* = 4), health problems such as high blood pressure or uncontrolled diabetes (*n* = 2), being misdiagnosed post-COVID (*n* = 1), or leaving the appointment (*n* = 3); another three participants were discarded at the end of the study because of their low MMSE score (see Figure 4). Table 1 shows the demographic characteristics of participant groups, their mean score on the MMSE test, and the elapsed time since their recovery from COVID-19 infection (ToD). Among the 109 post-COVID participants, some reported sequelae in the respiratory system (*n* = 47), cardiovascular system (*n* = 8), neurological system (*n* = 33), and other type of symptoms or discomforts (*n* = 50) including ageusia, anosmia, hair loss, fatigue, etcetera.

The *t*-test analysis between the Control and post-COVID group did not show a significant difference in age *t*(142) = −0.744 (*p* = 0.458). Once the post-COVID group was divided into Mild-Moderate and Severe-Critical groups, only the Severe-Critical group presented a significant difference in age versus the Control (*t*(76) = −2.666, *p* = 0.009) and Mild-Moderate (*t*(104) = −3.635, *p* < 0.001) groups. A Bartlett’s analysis confirmed that the groups had the same variance in their age distribution with a resulting Bartlett’s statistic *T*(142) = 2.561 (*p* = 0.278). The years of education in our sample did not reach a significant difference between groups after an analysis with a Wilcoxon’s test, where the comparisons were reported a *Z* = 0.296 (*p* = 0.767) for the CT-MM groups, *Z* = 0.599 (*p* = 0.549) for the MM-SC groups, and a *Z* = 0.799 (*p* = 0.424) for the CT-SC groups. A Levene’s quadratic analysis also confirmed the same variance in years of education between the groups with a statistic *W* (2, 141) = 1.566 (*p* = 0.212).

The 2AFC and SRT data were pre-processed for eliminating inconsistencies mentioned in the Quantitative Variables Section. Regarding repetitive choice patterns, the 2AFC test data from 86 participants (30 control; 56 post-COVID) were considered for analysis of phase 1, and data from 95 participants (26 control; 69 post-COVID) were used for phase 2. Regarding quality assessment for EEG signals, participants with lower than 80% signal of expected quality were discarded, and 99 participants were considered for the performance metrics analysis (31 control; 68 post-COVID). For the SRT test, participants with fortunate responses or missed targets in more than 80% were discarded and 134 participants (32 control; 103 post-COVID) were considered for the test analysis.

The Kolmogorov–Smirnov test was used to evaluate the normality of the data, finding that the Efficiency Index *d’* for both phases showed non-parametric distributions, and reaction times and values of the cognitive metrics had parametric distributions. Therefore, a Wilcoxon test (which works with non-parametric and parametric samples) was performed between the groups (control and post-COVID) as an independent variable; *d’*, SRT times, and cognitive metrics (Focus, Interest, and Engagement) as dependent variables; significance value was defined at 0.05. Additionally, a correlation analysis was performed between groups; *d’*, SRT times were used as independent variables; age and ToD were used as dependent variables.

### 3.2. Outcome Data

The results of decision-making processes based on visual sensitivity and visual attention (2AFC test), as well as the reaction times, are shown in Table 2. The estimation of these events was computed based on confidence intervals for the medians in each test, taking as reference the performance of the Control group. Therefore, the 2AFC phase 1 test (2AFC-P1) considers a *Med* = 3.28 (95%CI [2.9, 3.28]), the 2AFC phase 2 test (2AFC-P2) a *Med* = 1.435 (95%CI [0.11, 2.08]), and the Simple Reaction Time test (SRT) a *Med* = 313.695 ms (95%CI [293.222 ms, 340.449 ms]). Based on these intervals and the number of events for our selected cognitive outcomes (*d’* for the 2AFC tests and the reaction time for the SRT test), the prevalence of cognitive deficiency for the Severe-Critical group reports with values of 37.50% for the 2AFC-P1, 46.15% for the 2AFC-P2, and 62.16% for the SRT. In the case of the Mild-Moderate group, the prevalence found in cognitive deficiency for the 2AFC-P1 test was of 12.50%, 23.26% for 2AFC-P2 test, and 42.42% for SRT test.

### 3.3. Two-Alternative Forced Choice Analysis

The 2AFC test has two phases, the first has the lowest level of fog density, and the second has the higher level of fog density. *f* values (Equation (1)) were defined by the mean *d’* reached by a healthy people test group, non-considered in this work. At different values of *f*, the lower level has an *f* value where all participants reach the perfect *d’* (3.29), and higher level has an *f* value where the participants reach a mean *d’*= 1.19 (80% of correct responses).

The median of *d’* for CT and MM is 3.28 for 2AFC-P1. Practically, both groups reached the perfect *d’*; however, the SC group reached a median *d’* = 2.9, a lower index in 2AFC-P1. This can be seen graphically in Figure 5. The comparative results for 2AFC-P1 indicate that the MM group did not present a significant difference compared to the CT group (*Z* = −1.328, *p* = 0.167) with a small effect size (Cohen’s *d* = −0.47) in favor of the MM group, but a significant difference versus the SC group (*Z* = 2.738, *p* = 0.006) was found with a large effect size (Cohen’s *d* = 0.85) compared to the SC group. Furthermore, the CT and SC groups did not present a significant difference (*Z* = 1.645, *p* = 0.100) and showed a medium effect size (Cohen’s *d* = 0.52), again disfavoring the SC group.

The evaluation of the influence of age on the *d’* of adult and middle-aged SC and MM sub-groups showed that middle-aged participants of MM and SC groups presented a significant difference in their *d’* (*Z* = 2.119, *p* = 0.034) with a large effect size (Cohen’s *d* = 1.10) in favor of the MM group. Table 3 has the mean and median *d’* values for CT, MM, and SC age sub-grouped.

It was found that the age presents an interaction with *d’* for the CT group explaining 17.2% of the variability of the efficiency index (adjusted *R^2^* = 0.172, *F*(1, 28) = 7.01, *p* = 0.013) with a coefficient *β* = −0.025 (*t*(28) = −2.648, *p* = 0.013); this means that each one-year increment represents a decreasing of the efficiency index, a fact that is expected as a consequence of a reduction in the sensitivity in the visual system due to aging [35]. The most representative predictor for the *d’* value achieved by the Mild-Moderate group was the categorical variable Sex (*β* = −0.257, *t*(16) = −2.704, *p* = 0.016); however, the interaction Respiratory × Neurological (*β* = −0.733, *t*(16) = −4.794, *p* = 0.00) was also an important predictor; these interactions with an adjusted R^2^ = 0.77 explain the 77% (*F*(4, 16) = 7.65, *p* = 0.00) of variability in the efficiency index for this group. This means that subjects that presented respiratory and neurological sequelae are more likely to present a lower visual sensitivity in the test, where female participants have a lower sensitivity. In the case of the Severe-Critical group, it was found that there exists an influence by the variables Sex, Age, and Respiratory, Cardiovascular, and Neurological symptoms. From these variables, the 38.9% of variability for *d’* (*F*(5, 23) = 3.47, *p* = 0.009) is mainly explained by the Age (*β* = −0.071, *t*(16) = −2.466, *p* = 0.022) and the interaction Age × Neurological (*β* = 0.101, *t*(16) = 2.585, *p* = 0.017), which manifests that the persistence of a neurological sequela influences the sensitivity of the participant as their age increases.

The relationship between the efficiency index *d’* and the participants’ age was tested under a correlation analysis with a significance value of 0.05, obtaining a significant correlation for CT group (*R* = −0.448, *p* = 0.013) and SC group (*R* = −0.522, *p* = 0.002). Conversely, MM group (*R* = −0.33, *p* = 0.115) did not present a significant correlation between these variables. Assuredly, the visual system reduces its sensitivity because of aging, predicting a reduction in cognition performance [35]. Based on this fact, it is expected that the participants present a decline over the *d’* as their age increases, which could be identified through a negative slope computed with linear regression. To illustrate this expected behavior, Figure 6 shows the results from the regression for each group, where the computed slopes resulted in negative values for the corresponding lines of each group (CT’s *R^2^* = 0.248; MM’s *R^2^* = 0.092; SC’s *R^2^* = 0.37).

To evaluate if ToD affects *d’,* a correlation analysis was performed. Coefficients for both groups showed a non-significant correlation, and computed values were *R* = −0.226 (*p* = 0.287) and *R* = −0.219 (*p* = 0.229) for the MM and SC groups, respectively. Consequently, it might be possible to anticipate different levels of influence independently of the elapsed time from the patients’ recovery in this study. The linear regression behavior for the MM group (*R*^2^ = 0.024) tends to be represented as a horizontal line rounding a high value of *d’* independent to ToD, as can be seen in Figure 7. In particular, the SC group regression was marked with a line of negative slope (*R*^2^ = 0.078), demonstrating that this group can even present lower values of *d’* after long ToD (see Figure 7). It is important to keep in mind that the non-correlation between the value of *d’* and ToD prevents us from taking this behavior as the main representative of the SC group.

Figure 8 shows that the median values (and standard deviations) of cognitive metrics of the groups behaved similarly, the statistics of which, by groups, are summarized in Table 4. This is reinforced by the results where the comparative Wilcoxon’s test on each metric showed non-significant differences between the comparisons CT-MM, CT-SC, and MM-SC. The numeric results of these comparatives for 2AFC-P1 and 2AFC-P2 are concentrated in Table 5.

The efficiency index for the 2AFC second phase present a significant difference between the CT group (*Mdn* = 1.435) and the SC group (*Mdn* = 0.110), with a value of *Z* = 3.143 (*p* = 0.002) and a very large effect size (Cohen’s *d* = 1.31) disfavoring the SC group. Likewise, the comparative of the MM group (*Mdn* = 0.180) against the SC group presented a significant difference (*Z* = 2.717, *p* = 0.007) with a medium effect size (Cohen’s *d* = 0.78) in favor of the MM group. The CT and MM groups (*Z* = 1.253, *p* = 0.21) showed non-significant differences and presented a small effect size (Cohen’s *d* = 0.47), indicating a better performance by the CT group. The efficiency indexes achieved by each group can be seen in Figure 9.

The *d’* reached by the different age subgroups present significant differences between groups, as can be seen in Table 6. The CT and MM adult groups showed a difference of *Z* = 2.299 (*p* = 0.022) with a medium effect size (Cohen’s *d* = 0.77); the MM and SC adult groups showed a significant difference of *Z* = 2.658 (*p* = 0.008) with a large effect size (Cohen’s *d* = 0.92); and the CT and SC adult groups reached a difference of *Z* = 3.746 (*p* = 0.00) represented by a very large effect size (Cohen’s *d* = 1.99).

For phase 2, the *d’* of the CT group showed an interaction with both Age and Schooling variables, from which Age appears as the most significant predictor (*β* = −0.077, *t*(23) = −7.210, *p* < 0.001), then, maintaining Schooling as a constant, the Age explains 68.7% of the variability for *d’* (adjusted *R*^2^ = 0.687, *F*(2, 23) = 28.4, *p* < 0.001). Like phase 1, this result matches with the expected reduction of the visual sensitivity derived from aging. Similarly, the Mild-Moderate post-COVID group presented an interaction between *d’* and Age but with a different predictor, in this case Sex. Still, for this group Age (*β* = −0.069, *t*(39) = −3.726, *p* = 0.001) represents the most significant explanation for 33.8% of the variability (adjusted *R*^2^ = 0.338, *F*(2, 39) = 8.14, *p* < 0.001) for our dependent variable (*d’*) too. Therefore, the visual sensitivity experienced by this post-COVID group decreases as the age of the subject increases, which shows an independence from the COVID sequelae. In particular, the Severe-Critical group manifested post-COVID symptoms as predictors for the efficiency index, i.e., respiratory, cardiovascular, and neurological disorders. From these interactions, the Cardiovascular sequelae (*β* = 0.441, *t*(20) = 2.397, *p* = 0.026) distinguishes individually; however, the presence of both Respiratory and Neurological symptoms, Respiratory × Neurological (*β* = −0.442, *t*(20) = −2.182, *p* = 0.041), explains 19.9% of the variability of *d’* (adjusted *R*^2^ = 0.199, *F*(3, 20) = 2.24, *p* = 0.09). Consequently, the visual sensitivity in decision-making processes is more likely to be related to sequelae produced by the COVID infection for this group.

The coefficients of correlation are for the CT, *R* = −0.805 (*p* < 0.001), and for MM, *R* = −0.435 (*p* = 0.004). Here MM and CT groups showed a significant relationship between the participants’ *d’* and their age, while the SC group did not reach such significance with a value of *R* = 0.021 (*p* = 0.919), maybe due to their cognitive damage and the poor *d’* reached by all SC participants. Same to 2AFC-P1, the linear regression analysis of participants shows that *d’* decreases when age increases for the CT *R*^2^ = 0.7, and for the MM *R*^2^ = 0.365, with negative slope values, for the SC group *R*^2^ = 0.002, remain almost constant (with low *d’*). The linear regression for the participants’ average *d’* versus age can be seen in Figure 10.

The relationship between participants’ average *d’* and ToD for 2AFC-P2 test has a correlation coefficient for MM group *R* = 0.028 (*p* = 0.859) and for SC group *R* = 0.153 (*p* = 0.457), indicating almost a null correlation. The linear regression of *d’* shows a very small increase as the ToD increases, this can be seen in Figure 11. Considering phase 1 of the 2AFC, the SC group represents a tendency of poor decision making among the post-COVID participants; however, the non-correlation between the value of d’ and ToD casts doubt on this tendency as a representative lasting affectation of their cognitive process.

The cognitive metrics along 2AFC-P2 differed from the previous phase. Focus was lower for the SC group (*M* = 0.386, *SD* = 0.029, *Mdn* = 0.394), as can be seen in the boxes from Figure 12. Certainly, the results expose that this group presents a significant difference compared to both the CT group (*Z* = 2.473, *p* = 0.013) with a large effect size (Cohen’s *d* = 1.02) and the MM group (*Z* = 2.497, *p* = 0.013) with a large effect size too (Cohen’s *d* = 0.88). The statistical results achieved by the groups can be seen in Table 4.

### 3.4. Simple Reaction Time Analysis

The reaction times show an increasing pattern between groups, SC group presented the slowest reaction time (*M* = 367.75 ms, *SD* = 68.679 ms, *Mdn* = 360.146 ms), as can be appreciated in Figure 13. The Wilcoxon’s test revealed a significant difference between the SC group and CT group (*M* = 315.455 ms, *SD* = 39.843 ms, *Mdn* = 313.695 ms) with a result *Z* = −3.652 (*p* < 0.001) and a large effect size (Cohen’s *d* = −0.91), favoring the CT group. Identically, SC group presented a difference versus MM group (*M* = 333.553 ms, *SD* = 57.889 ms, *Mdn* = 327.57 ms) of *Z* = −2.767 (*p* = 0.006) and a medium effect size (Cohen’s *d* = −0.55), indicating slower reactions for the SC group. Contrarily, CT and MM groups did not present a difference, where the comparison resulted in a value of *Z* = −1.428 (*p* = 0.153) showing a small effect size (Cohen’s *d* = −0.34).

Based on the reaction times achieved by the adults and middle-aged subgroups (Table 7), the post-COVID groups remained to present significative differences between them and compared to the Control group, except for the comparisons between the adults of the CT and MM groups (*Z* = −0.400, *p* = 0.689) and the middle-aged of both post-COVID groups (*Z* = −0.278, *p* = 0.781); these results may represent little or no impairment in the reaction time for the adults, and a significative impairment for the middle-aged independently of the severity of the disease.

The adult SC group presented the lowest performance in SRT test, the comparison of MM and SC groups showed a significant difference *Z* = −2.748 (*p* = 0.006) with a medium effect size (Cohen’s *d* = −0.77); also, a significant difference was found between CT and SC groups with *Z* = −2.575 (*p*= 0.01) and a large effect size (Cohen’s *d* = −0.9). On the other hand, the middle-aged subgroups of the post-COVID participants had significant differences compared to the CT group, where the MM group presented a difference of *Z* = −2.09 (*p* = 0.037) and a medium effect size (Cohen’s *d* = −0.79); meanwhile, the SC group presented a difference of *Z* = −2.325 (*p* = 0.02) with a large effect size (Cohen’s *d* = −0.86).

The interactions found for the reaction time were Sex × Schooling for the control group (*β* = −10.465, *t*(28) = −1.823, *p* = 0.079), mainly caused by Sex influence (*β* = 195.18, *t*(28) = 2.199, *p* = 0.036), which explains 18.2% of the variability in the reaction times (adjusted *R*^2^ = 0.182, *F*(2, 28) = 3.3, *p* = 0.036); in this situation, females from the Control group are more likely to present slower reaction times than males. The 47.9% of variability for the reaction times for the Mild-Moderate group (adjusted *R*^2^ = 0.479, *F*(5, 55) = 6.98, *p* < 0.001) is explained by the interaction of the variables Sex, Age, Schooling and Respiratory and Neurological post-COVID disorders. From these predictors, Age is the most representative in the model with a coefficient *β* = 8.74 (*t*(55) = 3.612, *p* = 0.001), which indicates that the reaction time of the participants increased approximately 8.74 ms for each one-year increment. The reaction times of the Severe-Critical group were found to interact with Sex and Neurological sequelae (Sex × Neurological), whose coefficient *β* = 100.05 (*t*(33) = 2.325, *p* = 0.026) is related to 20.4% of the variability of the dependent variable (adjusted *R*^2^ = 0.204, *F*(2, 33) = 4.07, *p* = 0.015). Following this, we can interpret that the females that reported any type of neurological sequela might present a slower reaction time, and therefore a lower information processing speed.

Coupled with Welford, the reaction time tends to increase in function of aging [25]. Accordingly, the CT group demonstrates this expected behavior based on the resultant positive slope from linear regression, as can be seen in Figure 14. The coefficient of determination computed for this group was *R*^2^ = 0.099. Similarly, post-COVID groups showed positive slopes but with higher values. The coefficients of determination for these groups corresponded to *R*^2^ = 0.295 for the MM group and *R*^2^ = 0.087 for the SC group. An analysis of correlation showed a significant relation between the participants’ average reaction time and age for the MM group with a coefficient of correlation *R* = 0.446 (*p* = 0.00), while the CT group (*R* = 0.279, *p* = 0.123) and the SC group (*R* = 0.202, *p* = 0.23) showed a non-significant relation among these variables.

The relation between the participants’ average reaction time and the ToD for post-COVID groups does not have a significant correlation. In this analysis, the MM group had a coefficient of correlation of *R* = 0.157 (*p* = 0.209) and SC group’s coefficient had a value of *R* = −0.149 (*p* = 0.378). The linear regression for the reaction time and the ToD for these groups resulted in lines with a negative slope for the SC group and a quasi-horizontal line fixed at a central reaction time value independently of the value of ToD for the MM group, as is shown in Figure 15. As a result of the regression, the MM group had a resultant coefficient of determination *R*^2^ = 0.007, while the SC group had a coefficient *R*^2^ = 0.055. These behaviors mark the performance decrease in reaction time and in the processing information speed, for the SC group as the ToD increments; meanwhile, the MM group shows a “constant” value for the reaction time for all the recovery periods. Still, these tendencies must be taken with caution due to the absence of correlation between the reaction time and the ToD.

### 3.5. Analysis of the Post-COVID-19 Severe-Critical Group by Age Groups

At the beginning of the pandemic, the middle-aged population was found to be the most affected by the COVID-19 virus, presenting many severe-critical cases [36]. Therefore, it was found to be interesting to analyze whether there is a difference between adult and middle-aged participants in tasks of visual-based decision and simple reaction times. As shown in Table 1, the participants in the post-COVID Severe-Critical group for this study have a mean age of 43.65 years (*SD* = 9.20 years); after dividing the Severe-Critical group by adults and middle-aged participants, each subgroup presents an average age of 36.57 (*SD* = 6.57) and 51.33 (*SD* = 3.88) years, respectively.

The *d’* and reaction times reached by the adult group and the middle-aged group are presented in Table 8. The differences between the adult and middle-aged subgroups were compared with a Wilcoxon’s test for the 2AFC and SRT evaluations. The 2AFC-P1 comparison was marked by a significant difference between the subgroups with a *Z* = 2.936 (*p* = 0.003) and a large effect size (Cohen’s *d* = 1.12) disfavoring the middle-aged subgroup. Contrary to 2AFC-P1 test, the 2AFC-P2 test did not present a significant difference between the age subgroups, where *Z* = 0.130 (*p* = 0.897) and a very small effect size (Cohen´s *d* = 0.13). Similarly, the reaction times did not present a significant difference between the age subgroups with a *Z* = −0.319 (*p* = 0.75) and a small effect size (Cohen’s *d* = −0.35). These last analyses can be interpreted as a common diminution of the decision-making for both subgroups in tasks requiring a higher visual sensitivity and the processing information speed.

In previous analyses, it was found that the *d’* and the reaction times did not present a correlation with the recovery period. To discard any age confounding, data for the Severe-Critical group were age-corrected with an intercept method. This method was done using the equation $Y_{a}=Y_{0}\left(A_{a}-A_{i}\right)/\left(A_{0}-A_{i}\right)$, where $A_{a}$ is the adjusted ToD, $Y_{0}$ is the observed ToD of the participant, $A_{a}$ is the standard age (average age), $A_{i}$ is the (age) intercept of the regression line corresponding to zero recovery time, and $A_{0}$ is the observed age. After this correction, the *d’* and the ToD for the 2AFC-P1 test remained uncorrelated for the MM group (*R* = −0.233, *p* = 0.272) but showing a correlation for the SC group (*R* = −0.463, *p* = 0.008). Figure 16 presents the result of the linear regression for the MM (*R*^2^ = 0.054) and SC groups (*R*^2^ = 0.215) after correcting the data for the influence of age; the behavior of these lines is like the effects seen in the linear regression for data without age correction (see Figure 7) but with higher values of *R*^2^.

The correction of age influence for 2AFC-P2 also showed similar behavior for the linear regressions (Figure 17) as its uncorrected counterpart (Figure 11), presenting positive slopes and better *R*^2^ values (*R*^2^ = 0.028 for the MM group and *R*^2^ = 0.117 for the SC group). In addition, the correlation analysis between the efficiency index *d’* and ToD remained non-correlated but showed better correlation values for both groups with *R* = 0.168 (*p* = 0.282) for the MM group and *R* = 0.342 (*p* = 0.08) for the SC group.

Unexpectedly, after correcting the reaction times for age influence, the linear regression for the MM group (*R*^2^ = 0.03) showed a different behavior (showed in Figure 18) to its previous behavior with an uncorrected age effect (see Figure 15) by presenting a positive slope, which might be interpreted as a small increment of the reaction times as the recovery time increments, revealing a possible diminution in the information processing speed. However, correlation analysis showed again that reaction time and ToD still do not present a relation for any post-COVID group; the MM group has a correlation value *R* = 0.174 (*p* = 0.163) and the SC group presents a correlation value of *R* = −0.132 (*p* = 0.437). The linear regression for the SC group (R^2^ = 0.017), shown in Figure 18, maintained a similar behavior after the age correction, as seen in its uncorrected counterpart (see Figure 15).

### 3.6. Risk Estimates

The cognitive deficiency’s prevalence odd ratios (PR) obtained by the Mild-Moderate and Severe-Critical groups during the 2AFC phase 1 were *PR* = 0.536 and *PR* = 1.607, respectively. Correspondingly, during the phase 2 in this same test they were *PR* = 1.209 and *PR* = 2.4. Finally, during the SRT test, the Mild-Moderate group presented a *PR* = 1.508 and the Severe-Critical group a *PR* = 2.21.

## 4. Discussion

The present work aimed to evaluate the cognitive processes of decision-making by visual sensitivity, visual attention, and information processing speed. Based on our study, we confirmed our hypothesis that post-COVID patients who suffered a severe-critical infection showed a lower efficiency index in decision-making tasks demanding high sensitivity of the visual system in comparison to the Control group (*Z* = 3.143, *p* = 0.002), and the post-COVID participants who recovered from a mild-moderate infection, whose difference was *Z* = 2.717 (*p* = 0.007). The processing information speed, measured with the reaction time test, exposed that severe-critical post-COVID patients presented slower response times compared to never-infected subjects (*Z* = −3.652, *p* < 0.001) and mild-moderate post-COVID cases (*Z* = −2.767, *p* = 0.006). These results confirm our hypothesis that the resulting damage in skills was more severe in severe-critical patients than for the mild cases, compared to non-infected subjects.

We would like to emphasize that this kind of study became relevant after the appearance of the COVID-19 pandemic and would not have been considered otherwise. Our proposal of a single evaluation study is the most direct approach for the estimation of the resulting deficiencies, and there is no initial reference to the cognitive performance of the participants since no prediction of the possible effects was available at the beginning of the pandemic. That is, the visual response effects were not detected at the beginning of the pandemic; furthermore, the first effects reported were the taste and olfactory losses, which emphasizes that, if any initial studies would have been carried out at the beginning of the pandemic, they would have pointed in that direction. However, the advantage of only doing this study after two years of disease spreading is that the obtained results are more stable now than they would have been during intermediary epochs of the pandemic: it is important to notice that visual deficiencies have become longer term effects than taste or olfactory loses, which in most cases were recovered after time periods which depended on the subjects’ prior health conditions. Another variable which could be under consideration is the variety of (long-term) recovery times of each recruited participant, whose mean for subjects in this study was 8.70 months (*SD* = 5.04) for the mild-moderate cases, and 11.25 months (*SD* = 5.29) for severe-critical cases. We should consider, on the other hand, that a complete clinical analysis of the control participants was not evaluated, since their clinical history contemplated in the study was only based on a verbal interview. For example, we did not require any laboratory analysis to thoroughly verify the overall health of the patients, assuming this would not promote any risk of bias in the selection process. We should also mention that an unavoidable characteristic of the study is that the Severe-Critical post-COVID group has a higher mean age than the Control and Mild-Moderate groups, an effect consistent with the predominant global behavior of the disease, where most severe cases were reported among adults with a mean age above 40 years [8,9,36].

Previous studies suggest that COVID-19 may be the cause for cognitive deficiencies [16], a premise derived from the discovery of disturbances in the nervous and visual systems [3,37], which directly relate to several cognitive processes. As a result of our study, it is evident that the participants of the Severe-Critical group performed more poorly than the Control and Mild-Moderate post-COVID groups. The reduced functioning of cognitive processes in severe-critical could be determined from the evaluated loss of sensitivity of the visual system to discriminate stimulus in poorly contrasted environments. As has been reported [38], patients with a severe infection of COVID-19 were likely to develop ocular complications, such as alterations in the optic nerve and the visual cortex, which would be reflected as acute vision loss or even visual impairment [4,13,14,39]. The most outstanding result of our evaluation procedure became evident when performing the 2AFC phase 2 test: during the initial phase 1, it was evident that the SC group performed poorly compared to the other two groups, but the differences among MM and CT could be interpreted as contradictory results, since our results showed that the MM group performed equally or even better than the CT group during the 2AFC first phase as indicated by the small effect size favoring the MM group (*d* = −0.47). However, in the second phase the expected lower performance differences were confirmed even between age subgroups of the three different cases. We believe that this could be related to the existence of an unreported or unknown condition in some participants of the control group, which is related to our restricted clinical history of these participants. Another possibility may arise from post-pandemic lock-down effects on their vision, as reported by other studies [12,40]. However, we do not consider this to be the case in our study, since the participants’ working environments varied from home to office activities. We found that not only is the severity of the disease strongly related to the impairment of these cognitive processes, but there is also a persistent tendency among the older adult population to present this impairment independently of the severity of the disease.

The cognitive metrics used in this study allowed us to analyze the visual attention in both phases of the 2AFC test. During phase 1, all groups performed the test with similar awareness, showing no significant differences among them with respect to the Focus, Interest, and Engagement variables. On the contrary, phase 2 was marked by a significant difference of the CT and MM groups compared to the SC group, in particular for the Focus metric with large effect sizes, indicative of the damage to the visual system, with participants not being able to find the test visual signal. This assessment is confirmed by different studies which have found EEG abnormalities as possible consequences of neuronal damage provoked by the COVID-19 infection, as evidence of invasion of the CNS and PNS by this virus has become apparent [11]. These abnormalities have not only been reported on frontal and central changes in the EEG at rest and during high cognitive loads [10,11], but also with brain changes reflecting and atrophy of the crus II for a cognitive decline [41] and a possible relation with the ACE receptor [17]. However, some other factors and even treatment procedures (e.g., the prolonged periods of ICU treatment with long-term sedation) are also known as contributors to the impaired functioning of the brain.

The SRT test, which is widely used as a marker of the information processing speed and to study the general functioning of the CNS [25], showed that the SC group had the slowest reaction time among the groups, reflecting a lower functioning of the information processing speed. A relation of this cognitive process with COVID-19 was previously established for a risk estimator of death [18], and our study contributes extending this lower performance to severe-critical recovered patients. By considering the longest route available in the nervous system, going from the visual system to the CNS, and finalizing in a physical response by the foot, we might be able to uncover disturbances in the response of the motor system caused by neurological syndromes detonated by COVID-19 infection. These syndromes, mainly reported in hospitalized and severe-critical patients, include fibromyalgia, peripheral nerve damage, and critical myopathy [2,7,16]. In less severe cases, it has been reported that poorly organized movements are present in response to commands [6]. It was interesting to find that the adults recovering from a mild-moderate infection had similar reaction times to the Control group; meanwhile, the post-COVID middle-aged participants presented similar reaction times independently of the severity of the COVID-19 infection. This relation might uncover an influence of the age and the severity of the infection to hypothesize that adults preserve a good functioning of the information processing speed, while the severe-critical cases result in an amelioration of this ability which, on one hand, could be a consequence of the infection severity, e.g., as we report for the predictors sex and neurological sequelae, or on the other hand, the infection severity could be a consequence of a previous reduced functioning of the information processing speed, as found by Batty et al. [18].

The linear regressions of the 2AFC tests show a decreasing tendency of *d’* in its first phase, which was even more noticeable for the Severe-Critical group. In the case of the second phase, the post-COVID-19 groups maintained a quasi-constant *d’* independently of the elapsed time of being discharged; however, the Severe-Critical group kept a lower index than the Mild-Moderate group, which goes in line with the first phase of the test, recognizing a diminished visual sensitivity only in severe cases. In addition, it was found that *d’* was uncorrelated to the discharged period of the patients in both tests, thereby exposing an unclear tendency of the amelioration of decision-making based on visual sensitivity for post-COVID-19 patients. The SRT test also showed a small decreasing tendency in the regression of the relation between the reaction time and the elapsed time from the patient’s discharge for the Severe-Critical group, while the Mild-Moderate group showed a tendency of a constant value on the regression. Contrary to the 2AFC, this decrease may indicate an improvement in the information processing speed of the patients as the reaction times are faster. However, the reaction time and the period of discharge were uncorrelated; thus, this apparent improvement cannot be asserted.

Despite the simplicity of our study, we were able to evaluate the functioning of some cognitive processes such as the decision-making based on visual sensitivity, visual attention, and the information processing speed through forced choice and reaction times. Based on the evidence we found of disturbances in the selected cognitive processes, we find it interesting to extend this study of cognition in post-COVID patients to some other areas such as the memory, learning skills, executive functions, and other sensory-perceptual responses. This becomes relevant as these cognitive processes are strongly related to our daily tasks in different environments such as the home, the work, the school, or social areas, where a disfunction in cognition might represent a lower performance in our tasks and less awareness of our surroundings.

The functioning of cognitive processes in recovered COVID-19 patients can be assessed using visual psychophysics evaluations to measure the decision-making based on visual sensitivity, visual attention, and information processing speed through choice and reaction times. We found that not only is the severity of the disease strongly related to the impairment of these cognitive processes, but also there is a relation of some respiratory, cardiovascular, and neurological sequelae derived from the infection. A persistent tendency among the middle-aged population to present this impairment, independently of the severity of the disease, highlights the necessity of an adequate follow-up for these patients. Admittedly, as most of the authors agree, it would be relevant to offer an adequate follow-up, not only for this population, but for all patients recovering from COVID-19.

## 5. Conclusions

We proposed a study with a set of psychophysics experiments to investigate if recovered patients from COVID-19 present a diminution on their skills related to decision-making based on visual sensitivity and processing information speed. We found that as result of the sequelae derived from the COVID-19 infection, in comparison to never infected individuals or patients recovered from a mild-moderate infection, patients recovering from a severe-critical infection have a poorer performance in tasks requiring a high visual sensitivity to make a decision and a slower processing information speed, being both cases an effect of a damage either of the visual system or the nervous system. As mild-moderate cases were found to present a similar performance to never infected individuals in decision-making and processing information speed, it still concerning the prevalence of a cognitive deficit in the processing information speed, as it is present on 4 of 10 patients based on our analysis.

The psychophysics experiment proposed to evaluate the decision-making process based on visual sensitivity, we used two different levels of fog density, i.e., two different visual sensitivity thresholds, to evaluate the performance of the recovered patients from COVID-19. The lesser fog density, or the less difficult level in the experiment, was easily solved by the never-infected and the mild-moderate cases, meanwhile patients with a severe-critical infection presented a lower performance. Similarly, these patients also had a poorer performance in the task presenting a higher value of fog density than both mild-moderate cases and never-infected individuals. Also, for this level the mild-moderate cases presented a barely lower performance than never-infected individuals, which allows us to believe that our testing tool might help to identify the severity of the infection suffered by the infected people.

Finally, we find important to extend the study of the cognitive processes among the infected population with COVID-19, this because there exists other unexplored areas and processes such as the memory, learning skills, executive functions, and other sensory-perceptual responses. Above all, it is relevant to analyze the possibility of including a psychological follow-up for all the patients recovering from COVID-19 to prevent an identify cognitive impairments as these processes are very important in our daily life tasks.

## Figures and Tables

**Figure 1 brainsci-12-01258-f001:**
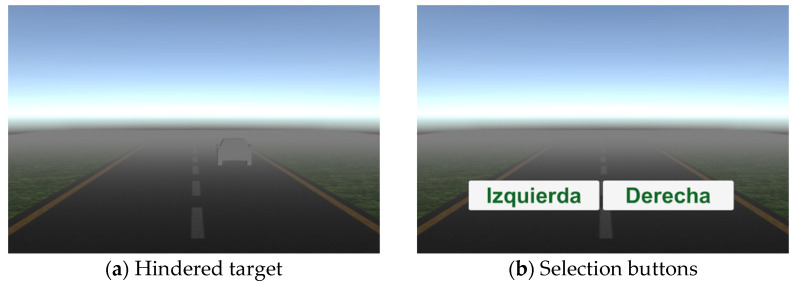
Two-Alternative Forced Choice (2AFC) test: (**a**) example of a hindered target with a low fog density; (**b**) choice buttons.

**Figure 2 brainsci-12-01258-f002:**
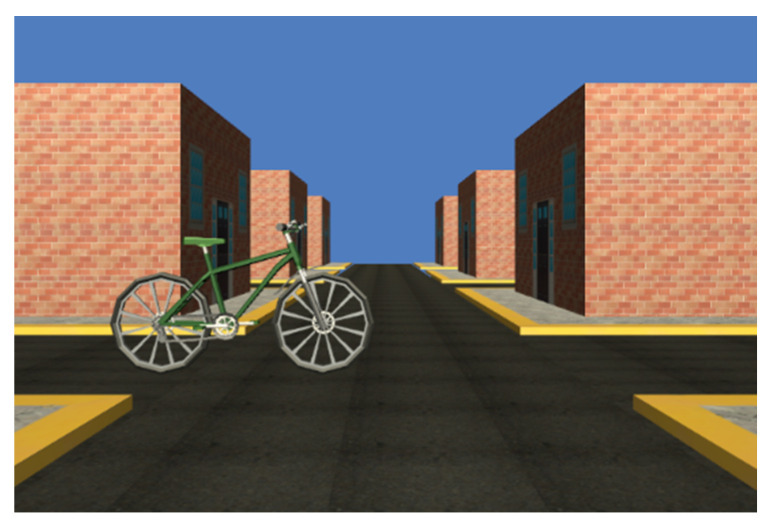
Bicycle in the VE programmed for Simple Reaction Time (SRT) test.

**Figure 3 brainsci-12-01258-f003:**
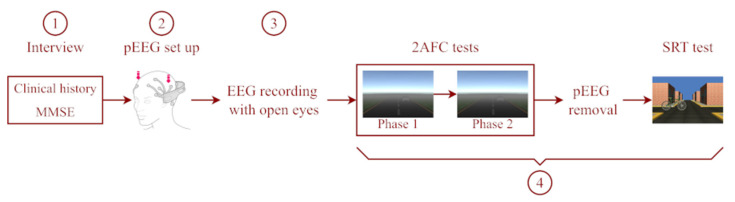
Steps of evaluation.

**Figure 4 brainsci-12-01258-f004:**
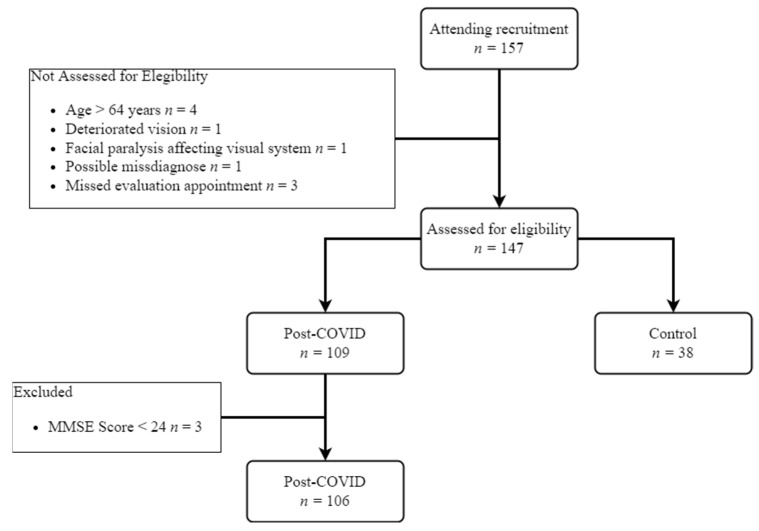
Recruitment and eligibility flowchart.

**Figure 5 brainsci-12-01258-f005:**
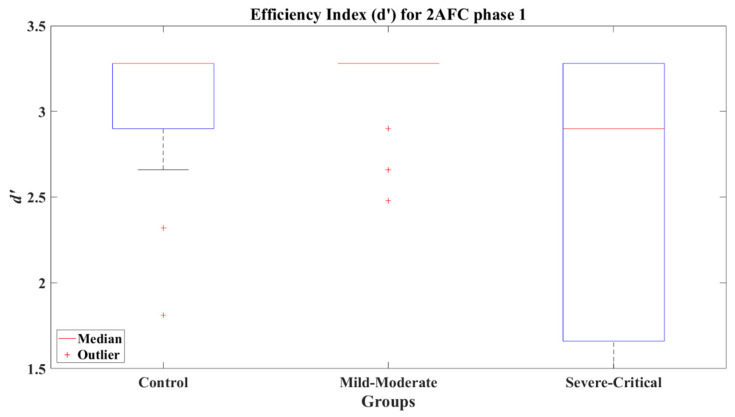
Efficiency index *d’* for 2AFC phase 1 (2AFC-P1) by group.

**Figure 6 brainsci-12-01258-f006:**
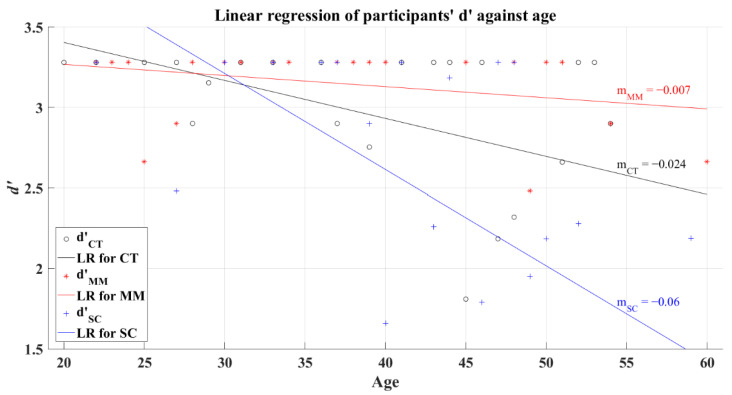
Linear regression of participants’ average *d’* versus age for 2AFC-P1. Black line—linear regression for Control group (LR for CT); red line—linear regression for Mild-Moderate post-COVID group (LR for MM); blue line—linear regression for Severe-Critical post-COVID group (LR for SC); $d{’}_{CT}$, Control group’s efficiency index; $d{’}_{MM}$, Mild-Moderate group’s efficiency index; $d{’}_{SC}$, Severe-Critical group’s efficiency index; $m_{CT}$, Control group’s slope; $m_{MM}$, Mild-Moderate group’s slope; $m_{SC}$, Severe-Critical group’s slope.

**Figure 7 brainsci-12-01258-f007:**
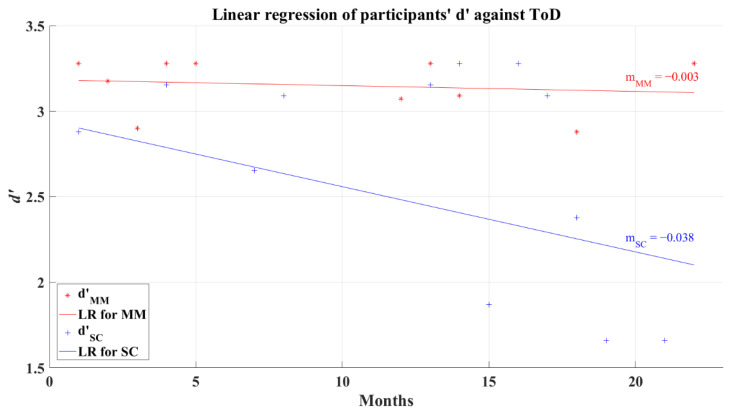
Linear regression for participants’ average *d’* and elapsed time of diagnosis for 2AFC-P1. Red line—linear regression for Mild-Moderate post-COVID group (LR for MM); blue line—linear regression for Severe-Critical post-COVID group (LR for SC); $d{’}_{MM}$, Mild-Moderate group’s efficiency index; $d{’}_{SC}$, Severe-Critical group’s efficiency index; $m_{MM}$, Mild-Moderate group’s slope; $m_{SC}$, Severe-Critical group’s slope.

**Figure 8 brainsci-12-01258-f008:**
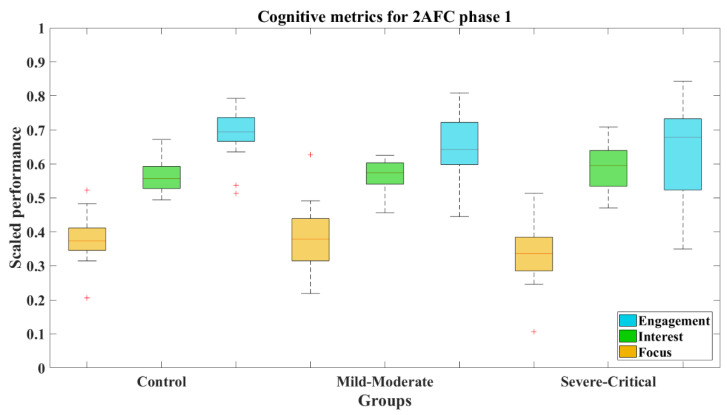
Cognitive metrics for 2AFC-P1 by group. Red line—median value of Focus, Interest, and Engagement for the corresponding group; red plus sign (+)—Outliers for the corresponding performance metric.

**Figure 9 brainsci-12-01258-f009:**
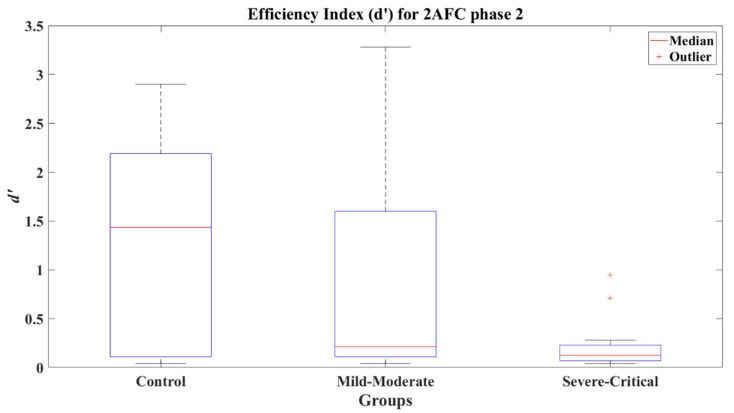
Efficiency index *d’* for 2AFC-P2 by group.

**Figure 10 brainsci-12-01258-f010:**
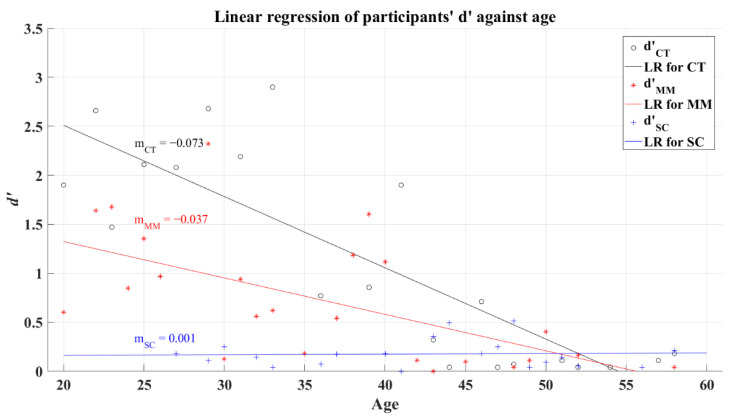
Linear regression of participants’ average *d’* versus age for 2AFC-P2. Black line—linear regression for Control group (LR for CT); red line—linear regression for Mild-Moderate post-COVID group (LR for MM); blue line—linear regression for Severe-Critical post-COVID group (LR for SC); $d{’}_{CT}$, Control group’s efficiency index; $d{’}_{MM}$, Mild-Moderate group’s efficiency index; $d{’}_{SC}$, Severe-Critical group’s efficiency index; $m_{CT}$, Control group’s slope; $m_{MM}$, Mild-Moderate group’s slope; $m_{SC}$, Severe-Critical group’s slope.

**Figure 11 brainsci-12-01258-f011:**
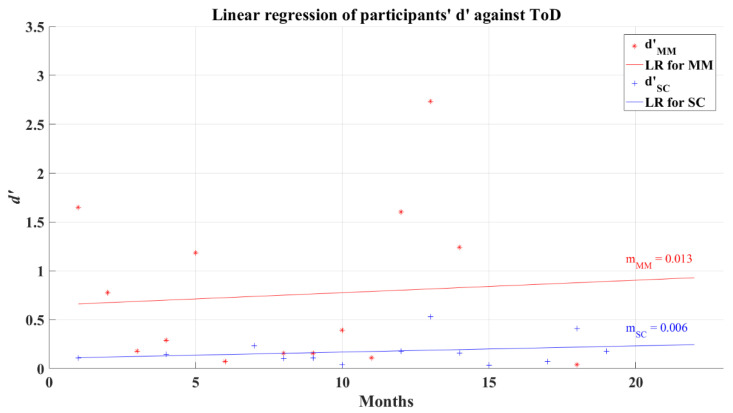
Linear regression for participants’ average *d’* and elapsed time of diagnosis for 2AFC-P2. Red line—linear regression for Mild-Moderate post-COVID group (LR for MM); blue line—linear regression for Severe-Critical post-COVID group (LR for SC); $d{’}_{MM}$, Mild-Moderate group’s efficiency index; $d{’}_{SC}$, Severe-Critical group’s efficiency index; $m_{MM}$, Mild-Moderate group’s slope; $m_{SC}$, Severe-Critical group’s slope.

**Figure 12 brainsci-12-01258-f012:**
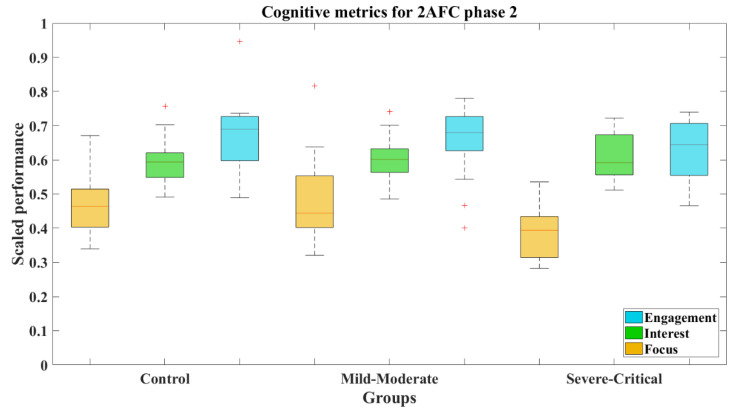
Cognitive metrics for 2AFC-P2 by group. Red line—median value of Focus, Interest, and Engagement for the corresponding group; red plus sign (+)—Outliers for the corresponding performance metric.

**Figure 13 brainsci-12-01258-f013:**
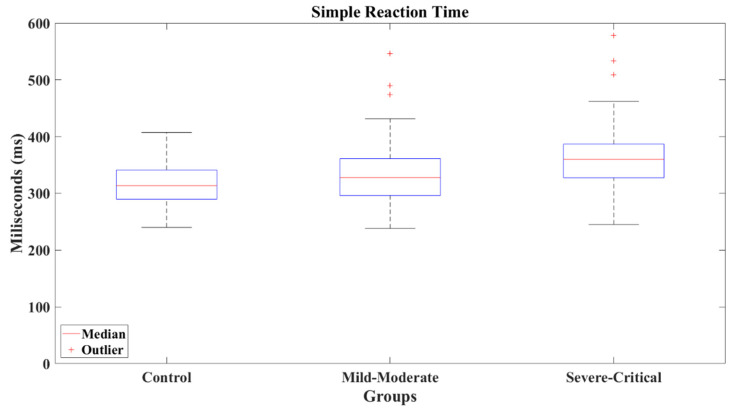
Simple reaction times by group.

**Figure 14 brainsci-12-01258-f014:**
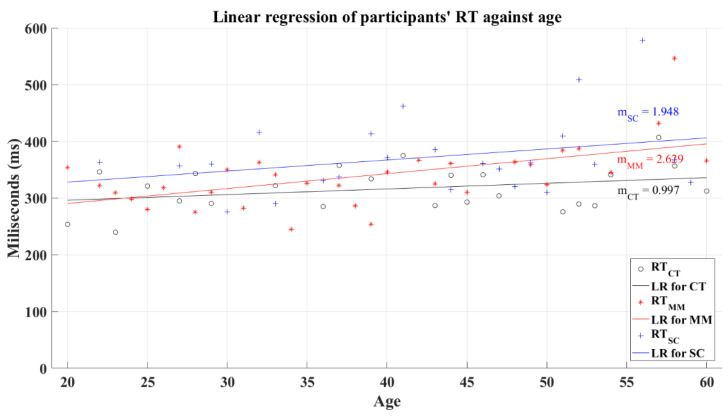
Linear regression for participants’ average reaction time versus age. Black line—linear regression for Control group (LR for CT); red line—linear regression for Mild-Moderate post-COVID group (LR for MM); blue line—linear regression for Severe-Critical post-COVID group (LR for SC); $RT_{CT}$, Control group’s reaction time; $RT_{MM}$, Mild-Moderate group’s reaction time; $RT_{SC}$, Severe-Critical group’s reaction time; $m_{CT}$, Control group’s slope; $m_{MM}$, Mild-Moderate group’s slope; $m_{SC}$, Severe-Critical group’s slope.

**Figure 15 brainsci-12-01258-f015:**
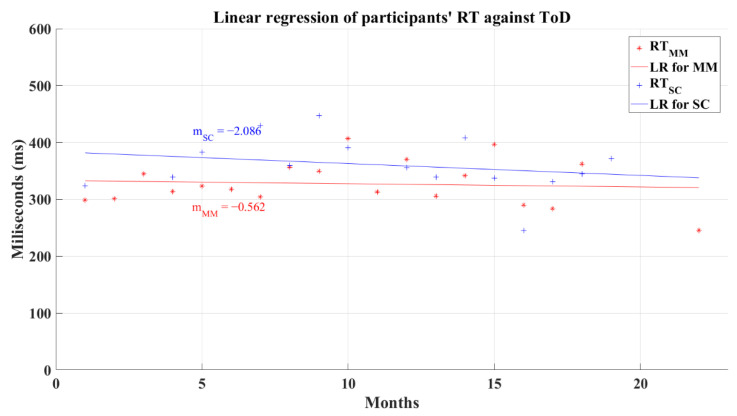
Linear regression for participants’ simple reaction time and elapsed time of diagnosis. Red line—linear regression for Mild-Moderate post-COVID group (LR for MM); blue line—linear regression for Severe-Critical post-COVID group (LR for SC); $RT_{MM}$, Mild-Moderate group’s reaction time; $RT_{SC}$, Severe-Critical group’s reaction time; $m_{MM}$, Mild-Moderate group’s slope; $m_{SC}$, Severe-Critical group’s slope.

**Figure 16 brainsci-12-01258-f016:**
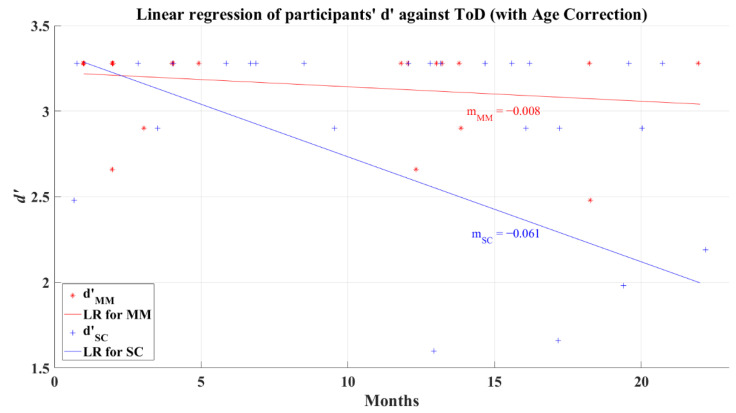
Linear regression of participants’ *d’* versus elapsed time of diagnosis with corrected age-effect for 2AFC-P1. Red line—linear regression for Mild-Moderate post-COVID group (LR for MM); blue line—linear regression for Severe-Critical post-COVID group (LR for SC); $d{’}_{MM}$, Mild-Moderate group’s efficiency index; $d{’}_{SC}$, Severe-Critical group’s efficiency index; $m_{MM}$, Mild-Moderate group’s slope; $m_{SC}$, Severe-Critical group’s slope.

**Figure 17 brainsci-12-01258-f017:**
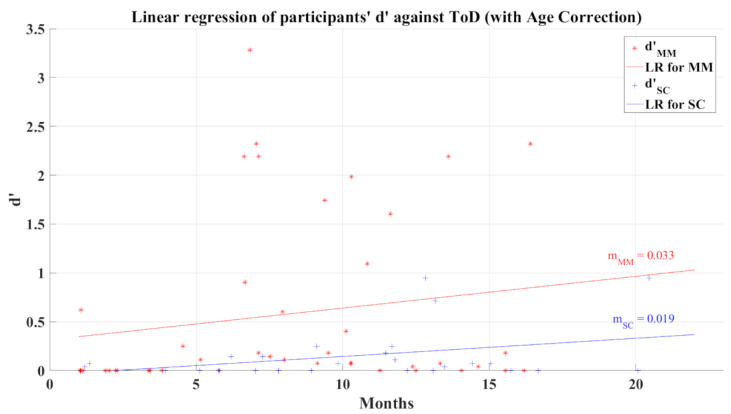
Linear regression of participants’ d’ versus elapsed time of diagnosis with corrected age-effect for 2AFC-P2. Red line—linear regression for Mild-Moderate post-COVID group (LR for MM); blue line—linear regression for Severe-Critical post-COVID group (LR for SC); $d{’}_{MM}$, Mild-Moderate group’s efficiency index; $d{’}_{SC}$, Severe-Critical group’s efficiency index; $m_{MM}$, Mild-Moderate group’s slope; $m_{SC}$, Severe-Critical group’s slope.

**Figure 18 brainsci-12-01258-f018:**
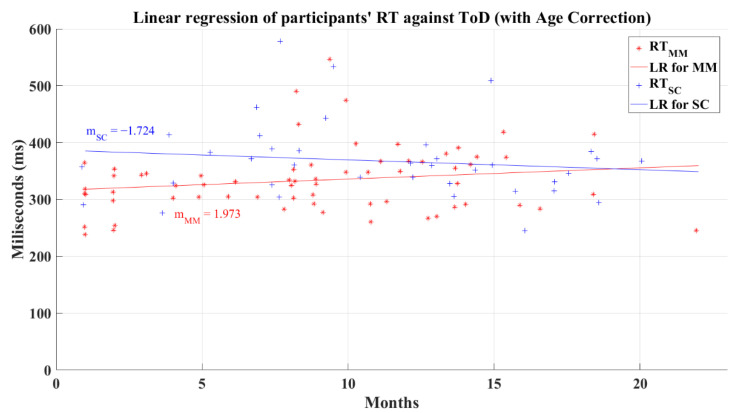
Linear regression for participants’ simple reaction time and elapsed time of diagnosis with corrected age-effect. Red line—linear regression for Mild-Moderate post-COVID group (LR for MM); blue line—linear regression for Severe-Critical post-COVID group (LR for SC); $RT_{MM}$, Mild-Moderate group’s reaction time; $RT_{SC}$, Severe-Critical group’s reaction time; $m_{MM}$, Mild-Moderate group’s slope; $m_{SC}$, Severe-Critical group’s slope.

**Table 1 brainsci-12-01258-t001:** Descriptive variables of the participants.

Group	Sex			Age Mean (*SD*)	Schooling	MMSE ^d^ Score (*SD*)	ToD ^e^ Months
	M ^a^	F ^b^	T ^c^		Years (*SD*)		
Control	25	13	38	37.29 (11.77)	15.00 (2.73)	29.00 (1.01)	-
Mild-Moderate	35	32	66	35.98 (11.24)	14.74 (3.09)	28.55 (1.43)	8.70 (5.04)
Severe-Critical	15	25	40	43.65 (9.20)	14.28 (3.49)	28.50 (1.13)	11.25 (5.29)

^a^ M: male participants; ^b^ F: female participants; ^c^ T: total of participants; ^d^ MMSE: Mini-Mental State Evaluation; ^e^ ToD: Elapsed time since being infected by COVID-19.

**Table 2 brainsci-12-01258-t002:** Events for cognitive outcomes.

Group	2AFC-P1 ^a^			2AFC-P2 ^b^			SRT ^c^		
	N ^d^	CD ^e^	No CD	N	CD	No CD	N	CD	No CD
Control	30	7	23	26	5	21	32	9	23
Mild- Moderate	24	3	21	43	10	33	66	28	38
Severe-Critical	32	12	20	26	12	14	37	23	14

^a^ 2AFC phase 1; ^b^ 2AFC phase 2; ^c^ Simple Reaction Test; ^d^ number of events (participants); ^e^ cognitive deficiency.

**Table 3 brainsci-12-01258-t003:** Values of *d’* for 2AFC phase 1 by age.

Age Group	Control		Mild-Moderate		Severe-Critical	
	*M* (*SD*)	*Med*	*M* (*SD*)	*Med*	*M* (*SD*)	*Med*
Adult	3.144 (0.268)	3.280	3.221 (0.171)	3.280	2.973 (0.618)	3.280
Middle-aged	2.537 (0.828)	2.780	3.023 (0.343)	3.280	2.011 (1.072)	1.980

**Table 4 brainsci-12-01258-t004:** Statistical results of performance metrics for 2AFC tests.

Cognitive Metric	2AFC-P1 ^a^			2AFC-P2 ^b^		
	CT ^c^	MM ^d^	SC ^e^	CT	MM	SC
Focus	*M* = 0.379 *SD* = 0.027 *Mdn* = 0.373	*M* = 0.385 *SD* = 0.037 *Mdn* = 0.379	*M* = 0.333 *SD* = 0.016 *Mdn* = 0.336	*M* = 0.473 *SD* = 0.063 *Mdn* = 0.464	*M* = 0.479 *SD* = 0.044 *Mdn* = 0.445	*M* = 0.386 *SD* = 0.029 *Mdn* = 0.394
Interest	*M* = 0.565 *SD* = 0.023 *Mdn* = 0.557	*M* = 0.569 *SD* = 0.019 *Mdn* = 0.574	*M* = 0.590 *SD* = 0.021 *Mdn* = 0.595	*M* = 0.594 *SD* = 0.022 *Mdn* = 0.594	*M* = 0.600 *SD* = 0.024 *Mdn* = 0.601	*M* = 0.610 *SD* = 0.027 *Mdn* = 0.592
Engagement	*M* = 0.693 *SD* = 0.012 *Mdn* = 0.694	*M* = 0.650 *SD* = 0.032 *Mdn* = 0.643	*M* = 0.624 *SD* = 0.016 *Mdn* = 0.678	*M* = 0.672 *SD* = 0.026 *Mdn* = 0.690	*M* = 0.664 *SD* = 0.020 *Mdn* = 0.679	*M* = 0.624 *SD* = 0.025 *Mdn* = 0.645

^a^ 2AFC phase 1; ^b^ 2AFC phase 2; ^c^ Control group; ^d^ Mild-Moderate group; ^e^ Severe-Critical group.

**Table 5 brainsci-12-01258-t005:** Comparative of cognitive metrics between groups.

Cognitive Metric	2AFC-P1 ^a^			2AFC-P2 ^b^		
	CT-MM ^c^	MM-SC ^d^	CT-SC ^e^	CT-MM	MM-SC	CT-SC
Focus	*Z* = 0.050 *d* = −0.07	*Z* = 1.561 *d* = 0.55	*Z* = 1.840 *d* = 0.59	*Z* = 0.400 *d* = −0.05	*Z* = 2.497* *d* = 0.88	*Z* = 2.473 * *d* = 1.02
Interest	*Z* = −0.650 *d* = −0.07	*Z* = −1.110 *d* = −0.40	*Z* = −1.335 *d* = −0.42	*Z* = −0.538 *d* = −0.09	*Z* = −0.197 *d* = −0.17	*Z* = −0.603 *d* = −0.24
Engagement	*Z* = 1.550 *d* = 0.45	*Z* = 0.277 *d* = 0.21	*Z* = 1.222 *d* = 0.59	*Z* = −0.041 *d* = 0.08	*Z* = 1.437 *d* = 0.43	*Z* = 1.226 *d* = 0.49

* *p* < 0.05; ^a^ 2AFC phase 1; ^b^ 2AFC phase 2; ^c^ Control versus Mild-Moderate; ^d^ Mild-Moderate versus Severe-Critical; ^e^ Control versus Severe-Critical.

**Table 6 brainsci-12-01258-t006:** Efficiency index *d’* for 2AFC phase 2 by age subgroups.

Age Group	Control		Mild-Moderate		Severe-Critical	
	*M* (*SD*)	*Med*	*M* (*SD*)	*Med*	*M* (*SD*)	*Med*
Adult	1.721 (0.975)	1.900	0.975 (0.967)	0.620	0.216 (0.278)	0.125
Middle-aged	0.163 (0.227)	0.090	0.137 (0.106)	0.125	0.181 (0.252)	0.090

**Table 7 brainsci-12-01258-t007:** Reaction times by age subgroups (presented values are in milliseconds).

Age group	Control		Mild-Moderate		Severe-Critical	
	*M* (*SD*)	*Med*	*M* (*SD*)	*Med*	*M* (*SD*)	*Med*
Adult	312.957 (40.196)	318.002	318.329 (46.152)	315.769	356.080 (55.770)	360.181
Middle-aged	320.952 (40.610)	308.438	368.566 (67.589)	359.028	380.068 (79.889)	355.642

**Table 8 brainsci-12-01258-t008:** Severe-critical group statistics by age subgroups.

Subgroup	2AFC-P1 ^a^	2AFC-P2 ^b^	SRT ^c^
Adult	*M* = 2.973 *SD* = 0.618	*M* = 0.216 *SD* = 0.278	*M* = 356.080 ms *SD* = 55.770 ms
Middle-aged	*M* = 2.011 *SD* = 1.072	*M* = 0.181 *SD* = 0.252	*M* = 380.068 ms *SD* = 79.889 ms

^a^ 2AFC phase 1; ^b^ 2AFC phase 2; ^c^ Simple Reaction Time test.

## Data Availability

The data obtained from the Two-Alternative Forced Choice (participants’ selections for both phases) and Simple Reaction Time (reaction times) are available. The data of this study can be found here: https://drive.google.com/drive/folders/1ZbxD7x-t0BzON7ki_ZEB_pGfAvnQKWg9?usp=sharing (accessed 16 July 2022).

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
