# Peer review of "COVID-19 Long-Term Effects: Is There an Impact on the Simple Reaction Time and Alternative-Forced Choice on Recovered Patients?"

_brainsci, 2022, doi:10.3390/brainsci12091258_

Round 1

Reviewer 1 Report

The article concerns a current, interesting and important topic. It is clearly written, with a well-described review of the latest literature on the topic in question. The developed research method is clearly described and the applied EEG registration equipment (Emotive Epoc) is appropriate for this kind of research.

However, I have some comments and questions concerning method and statistical analysis, that should be considered and answered:

1) The weak point of the presented study is the lack of the initial tests of cognitive performance (CP) for all COVID-19 patients, e.g. 1 or 2  months after recovery, to observe CP’s changes over time. Such results would show whether cognitive performance after COVID-19 remaining at the same level after a long period of time or whether it is increasing / decreasing over the time.

 2) The obtained research results do not allow to state whether there is a long-term influence on the performance. The results only show a certain level of CP at the certain moment of time after the recovery, but without reference to the earlier CP.

 3) Subjects could have had health problems affecting their CP both before they were infected with SARS-Cov-2 and after recovery, but before the participation in the study (this period was quite long and exceeded even 12 months). This is especially important in the SC group with older patients. Have the examined patients suffered from other diseases that could affect cognitive performance in the period before and after COVID-19? Whether there was information about it obtained from the subjects when they were classified for research?

4) Taking into account the different size of groups and age differentiation, the studied groups were not homogeneous. As a result, this inhomogeneity should be carefully included in the statistical analysis. The presented results of  correlation analysis against age are ambiguous as there is no separation of the influence of age and the influence of the COVID-19 on cognitive performance. Correlation analysis with controlling the influence of age on CP should be carried out.

 5) "Linear regression for participants’ ………     and elapsed time of diagnosis…” (figure: 6, 10 and14) are not convincing.  They presents different subjects’ efficiency index d’ and reaction time.

 6) The limitations of the carried out study should be written in the article and included in the conclusions.

The noticed mistakes or inaccuracies in the content of the article are presented below:

 d’ is differently named in the text: 1) efficiency index (lines: 140, 292, 307, 364) or 2) performance index (line 175).

·         Formula 2, there is Z(F) instead of Z(A)

·         Line 343: there is t-tests instead of Wilcoxon’s test

·         Line 630, Reference position nr 31 is not complete.

The article needs major revision.

Author Response

We extend our best regards, and we would like to thank you for your commentaries and observations about our presented research paper. We appreciate them as they allowed us to improve our manuscript in a great manner. In general, the manuscript has received major changes in accordance with the enriching commentaries received by the reviewers, from which we would like to stand out the following:

  • The manuscript has been adapted to follow the STROBE guidelines to ensure that our study comprehends the key points of a cross-sectional study. We include the checklist for the manuscript. Following this change the manuscript is clearer in relation to the study design, the characteristics of the target population of study, the outcomes used and their relationship with our objectives and hypotheses, the presentation of the results and their interpretation with a clearer discussion.
  • The title of the work has changed to specify which abilities from the broad aspects of the cognition are we focused on. The title has been redefined as “COVID-19 Long-term Effects: Is There an Impact on the Simple Reaction and Choice Times of Recovered Patients?”.
  • The results section has been extended to include and mention the prevalence of the cognitive deficiency in post-COVID patients (Section 3.2 Outcome data, lines 422-437) when compared to our control group, including the risk estimates in section 3.6 lines 650-655. It is also included a set of analysis related to age influence in our study: first, in the Section 3.5 from line 583 to 605, we analyzed the severe-critical group by age groups (adult and middle-aged) to identify the impact of the COVID-19 and their differences in these subgroups; later, in the same section from lines 607 to 648, we adjusted the results for the post-COVID groups (mild-moderate and severe-critical) for age influence based on the intercept method.
  • The confounders and limitations of the study has been identified and added to the manuscript, which allowed us to rewrite a better understanding of our findings and to allow the reader acquiring a cautious interpretation of our results.
  • It was identified an error for the statistical description for the post-COVID and control groups in the Simple Reaction Time. The control group changed from (M = 330.41ms, SD = 55.025ms, Mdn = 313.695ms) to (M = 315.455ms, SD = 39.843ms, Mdn = 313.695ms) in lines 544-545; the mild-moderate post-COVID group changed from (M = 334.523ms, SD = 57.995ms, Mdn = 327.57ms) to (M = 333.553ms, SD = 57.889ms, Mdn = 327.57ms); the severe-critical post-COVID group changed from (M = 374.083ms, SD = 70.762ms, Mdn = 360.146ms) to (M = 367.75ms, SD = 68.679ms, Mdn = 360.146ms). These errors derived from a mistake reading the results from the recruited participants instead of the included for the study. This change did not influence the comparison analysis between groups because these comparisons were computed exclusively for the included participants. After this unpleasant mistake, we also verified and confirmed that the correct values of the included participants were taken and written in the manuscript for the 2AFC test in both phases. We extend an apology for this unfortunate mistake.
  • Finally, we would like to make of your knowledge that the data from the included participants can be found here: https://drive.google.com/drive/folders/1ZbxD7x-t0BzON7ki_ZEB_pGfAvnQKWg9?usp=sharing. This reference is indicated also in the document in lines 794-795.

 In attention to your personal contributions, we include a table with the corresponding answers to each one of the extended commentaries. Again, we appreciate your observations, and we thank you for sharing your experience to improve our manuscript.

Reviewer 2 Report

Work presented deals with the issue of cognitive impairments in adults affected by COVID-19 infection.

The work is well written and documented, only few aspects remain to be addressed:

  • it would be interesting to observe the impact of cognitive impairment in patients with severe-critical disease by dividing the group into age groups;

  • double-check the citations in the results as some are missing;

  • are data available on any follow-up?

Author Response

(The authors gave the same response as above.)

Reviewer 3 Report

Congratulations to the authors for this research work, but the manuscript requires a number of important changes and corrections. The authors of the article set the following objective: To determine the long-term effects on patients diagnosed with Covid-19 with reference to cognitive processes.

The article requires an in-depth review of the English language by an expert.

My comments and suggested changes to the article are as follows:

- The title is not well defined, be more specific and indicate in it the type of study you are doing and exactly what cognitive processes you are doing the intervention on.

- Rewrite the abstract should state the objective or purpose of the study, and contain a well-defined section for each of the sections of the manuscript. It should provide data on the results and the conclusions obtained should be well defined.

- Indicate keywords according to your manuscript you have not performed any type of virtual reality intervention, you should use MeSH terms according to your manuscript. Please select 4 to 6 words or short phrases that represent the essential content of the article.

- Because this text does not present bibliographic justification in the text: "All the neurological manifestations were found in both the CNS and the peripheral nervous system (PNS) with different degrees of severity and as para and post-infection stages."

- Because this expression needs 8 bibliographic references for its justification in the text, it is excessive: "These manifestations were mainly reported as cerebrovascular events such as strokes, intracranial hemorrhages, and vasculitis; alterations of the mental state manifested as encephalopathies, encephalitis, confusion, and delirium; some syndromes related to the PNS such as the Guillain-Barré syndrome [2,5,7–12]".

They should rewrite their introduction in accordance with the sources they consult.

- This is one of the most serious errors in his work, which is not well defined in terms of outcomes, since the term cognitive processes encompasses a multitude of processes such as sensory-perceptual aspects, attention, information processing, memory, thinking, executive functions, learning, language, creativity or motivation. You are using two tools to assess an aspect that is too broad and complex, which may pose a risk of bias.

Forced choice of two alternatives is a method to measure the sensitivity of a person, child or infant, or animal to some particular sensory stimulus, through the pattern of choices and response times of that observer to two versions of the sensory stimulus, I believe it is not an entirely appropriate tool to assess all cognitive processes (not stated in limitations of the study, there is high risk of bias and that your results are inconclusive).

Like the Simple Reaction Time Test (SRT), which measures information processing speed but not all cognitive aspects, the SRT measures the speed of information processing but not all cognitive aspects.

This should be clarified and reformulated in both its introduction and discussion.

- Adapt your manuscript to STROBE recommendations and attach a STROBE checklist for your type of study.

- How was it determined that the subjects in the control group had not suffered from Covid-19? Who determined the state of cognitive impairment or sequelae? Who evaluated this selection process? Don't you think it could be a bias in the study to include subjects from 20 to 60 years of age, since the cognitive level is quite different in subjects aged 20 than in those aged 60? Why 20 and not 18?

Also indicate the inclusion and exclusion criteria are not defined.

- In the methodology section, the first section should be about the design of the study and the type of recommendations or guidelines that have been followed for the execution of the work, since we do not know the type of study you are conducting, are we talking about a clinical or descriptive trial?

- The sample size is very small. Why is the number of participants so small? How did you solve this problem? How was the sample size determined?

- The section "Study Variables" is erroneous and should clearly define in a section the outcomes analyzed and the measuring instruments for the analysis of these variables, what the tests consisted of, scores, description of the scores.

- You must indicate the system or program with which you performed the data analysis (data analysis).

- Specify and clarify how they were performed: Randomization, type of randomization, mechanism used to implement the randomization sequence, who generated the randomization sequence, who selected the participants, and who assigned the participants to the interventions, masking.

- They must define the abbreviations entered in the tables in a legend.

- Discussions should cover the key findings of the study: discuss any previous research related to the topic to place the novelty of the discovery in the appropriate context, discuss possible shortcomings and limitations in its interpretations, discuss its integration into the current understanding of the problem and how. This advances current views, speculates on the future direction of research, and freely postulates theories that could be tested in the future, completed, and reformulated. The discussion needs to be rewritten, is sparse, inadequately justified and contains serious errors in the introduction of references.

- The authors have not provided a limitations section when the manuscript has major limitations (the results cannot be taken with such reliability since the main limitation of correlational studies is that the results do not indicate whether there is a cause-effect relationship between the variables considered and they are comparing subjects with and without pathology), the conclusions section is not defined in a clear and well defined way, the same references are used for the introduction and discussion of the results, see 2,3,7,10,11,12,13,14,16,17,20, this leads to the conclusion that the discussion has not been carried out according to the results obtained.

- It is recommended that authors review the journal's author guidelines regarding bibliographic references, which indicate that the name of the journal should appear in italics, and review some other formatting errors.

Author Response

(The authors gave the same response as above.)

Round 2

Reviewer 1 Report

Thank you for the carefully prepared answers and for making many profound changes in the article. However, I am  sorry to say that the further improvement of this article is needed.

Main reservations and comments

Due to the limitations of the study, resulting from the specifically designed research process, i.e. testing cognitive abilities, only after COVID (postCOVID) has been acquired, there can be justified doubts as to the reliability of the results. In a study designed in this way, the results are influenced by numerous confounding factors, including, in particular, the individual characteristics of the respondents. The influence of these factors on cognitive abilities tested by virtual environment tests is well known in the literature on the subject (e.g. age, level of education, computer literacy, ability to immerse in a virtual environment, gender). Additionally, the results of the analyzes presented in the article confirm the disturbing influence of these factors on the obtained results, e.g.:

·         line 378-381 "Once the post-COVID group was divided as mild-moderate and severe-critical groups, only the severe-critical group presented a significant difference in age versus the control (t (78) = -2.838, p = .006) and mild-moderate (t (107) = -3.733, p = .00) groups. "

·         age group differences in Table 1;

·         differences in the results of the dependent variable d' according to the age of the respondents and the degree of disease incidence presented in Figures 6, 7, 10.

All this means that one can have significant doubts whether the obtained results are more the effect of the influence of confounding factors, including, in particular, the age of the respondents, or the actual impact of complications after recover from COVID-19 .

Additional analyzes presented in the revised article are not convincing enough, as there is still no proper test of the effect of interaction between age, post-COVID complications and the results of dependent variables, which is necessary in analyzing the results of a study designed in this way.

The article lacks conclusions.

Additional editorial remarks

Lines 430-437: for both the MM and SC groups there are the same median values and ranges of variations of reaction time. In paragraph 3.4, lines 541-547, the values of the median of reaction time are probably correct: MM group - 327.57ms, SC group -360.146 ms.

Line 358 ; the text should show in the first sentence which parameter is discussed for 2AFC second phase.

Figures 7, 9, 11, 14, 15 please write explanation under the figures, what the symbols mSC, mCT and mMM  on the charts mean.

Suggestions for Authors

In order to obtain real evidence of long-term influence on the performance of recovered patients, it would be advisable to conduct an additional study of the same subjects. Only these results in comparison with the previous ones could be a proof of long-term influence on the performance.

The proper statistical test of the effect of interaction between age, post-COVID complications and the results of dependent variables is needed.

Author Response

We extend our best regards, and we would like to thank you for your new commentaries and observations about our research paper after the previous corrections. We appreciate them as they allowed us to improve even more our manuscript in a great manner. In general, the manuscript has received some changes in accordance with the enriching commentaries received by all the reviewers, 

Reviewer 3 Report

Although you have solved some errors, there are still several errors and unresolved issues that have not been corrected or your modifications have been overlooked in your responses. I suggest that you carefully read the recommendations requested in the first revision and solve all the errors indicated by providing a response in the manuscript or major changes in the text.

Author Response

(The authors gave the same response as above.)
